# Multiorgan and Vascular Tropism of SARS-CoV-2

**DOI:** 10.3390/v14030515

**Published:** 2022-03-03

**Authors:** Cédric Hartard, Ahlam Chaqroun, Nicla Settembre, Guillaume Gauchotte, Benjamin Lefevre, Elodie Marchand, Charles Mazeaud, Duc Trung Nguyen, Laurent Martrille, Isabelle Koscinski, Sergueï Malikov, Evelyne Schvoerer

**Affiliations:** 1Laboratory of Virology, Department of Microbiology, CHRU Nancy, Université de Lorraine, F-54000 Nancy, France; c.hartard@chru-nancy.fr; 2CNRS, LCPME, Université de Lorraine, F-54000 Nancy, France; ahlam.chaqroun@univ-lorraine.fr; 3Department of Vascular & Endovascular Surgery, CHRU Nancy, Université de Lorraine, INSERM 1116, F-54000 Nancy, France; n.settembre@chru-nancy.fr (N.S.); s.malikov@chru-nancy.fr (S.M.); 4Department of Biopathology, Centre de Ressources Biologiques, Université de Lorraine, F-54000 Nancy, France; g.gauchotte@chru-nancy.fr; 5Department of Infectious and Tropical Diseases, CHRU Nancy, Université de Lorraine, F-54000 Nancy, France; b.lefevre@chru-nancy.fr; 6APEMAC, Université de Lorraine, F-54000 Nancy, France; 7Department of Legal Medicine, CHU Nancy, Université de Lorraine, F-54000 Nancy, France; e.marchand2@chru-nancy.fr (E.M.); laurent.martrille@chu-montpellier.fr (L.M.); 8Department of Urology, CHRU Nancy, Université de Lorraine, F-54000 Nancy, France; c.mazeaud@chru-nancy.fr; 9Service ORL, CHRU Nancy, Université de Lorraine, F-54000 Nancy, France; dt.nguyen@chru-nancy.fr; 10Laboratoire de Biologie de la Reproduction/CECOS Lorraine, CHRU Nancy, INSERM U1256 NGERE, F-54000 Nancy, France; koscinski.isa@gmail.com

**Keywords:** SARS-CoV-2, viral quasispecies, ultra-deep sequencing

## Abstract

Although the respiratory tract is the main target of SARS-CoV-2, other tissues and organs are permissive to the infection. In this report, we investigated this wide-spectrum tropism by studying the SARS-CoV-2 genetic intra-host variability in multiple tissues. The virological and histological investigation of multiple specimens from a post-mortem COVID-19 patient was performed. SARS-CoV-2 genome was detected in several tissues, including the lower respiratory system, cardio-vascular biopsies, stomach, pancreas, adrenal gland, mediastinal ganglion and testicles. Subgenomic RNA transcripts were also detected, in favor of an active viral replication, especially in testicles. Ultra-deep sequencing allowed us to highlight several SARS-CoV-2 mutations according to tissue distribution. More specifically, mutations of the spike protein, i.e., V341A (18.3%), E654 (44%) and H655R (30.8%), were detected in the inferior vena cava. SARS-CoV-2 variability can contribute to heterogeneous distributions of viral quasispecies, which may affect the COVID-19 pathogeny.

## 1. Introduction

SARS-CoV-2, causing coronavirus disease 2019 (COVID-19), was detected for the first time in December 2019. Since then, this new pathogen has rapidly spread, with more than 300 million of infected people worldwide and more than 5 million of deaths. The clinical signs involve mainly the respiratory tract and the subsequent complication results in an acute respiratory distress syndrome (ARDS). As the COVID-19 pandemic continues to progress, knowledge about its wide-spectrum clinical manifestations and associated complications is advancing. Other organs failures are thus reported, such as cardiovascular involvement, with the description of myocarditis, vascular endothelial injury or thromboembolic events [1,2,3].

The ubiquitous distribution of the ACE2 receptors in several tissues may explain the multi-organ dysfunction linked to COVID-19, as well as the inflammatory and immune components [4]. Hence, the presence of the SARS-CoV-2 genome has already been reported outside the respiratory tract, most notably in the kidney, gastrointestinal system, nervous system and blood vessels, with some arguments in favor of a local replication [5,6,7,8,9,10,11].

To better understand the reasons for the wide-spectrum clinical features, it is essential to investigate the viral extrapulmonary dissemination of SARS-CoV-2. Thus, autopsy studies can be particularly relevant in order to characterize possible viral- and/or host-related characteristics linked to differential tissue affinity. In the present study, we explore this issue via multiple post-mortem histological and viral analyses, including ultra-deep sequencing, from specimens collected from a COVID-19 patient during the second wave of the pandemic (October 2020, France) in order to study intra-host variability.

## 2. Materials and Methods

### 2.1. Clinical Data and Autopsy

A 70-year-old man was admitted to Nancy University Hospital in October 2021. He presented a medical history of chronic heart failure due to atrial fibrillation and coronary disease, peripheral vascular disease with stroke and cognitive sequelae, arterial hypertension and active smoking. SARS-CoV-2 infection, confirmed by nasopharyngeal RT-PCR, began on 12 October 2021 with confusion, cough and pharynx erythema. After 6 days, he was admitted for confusion, dehydration and ARDS, associated with a typical aspect of lung infection on a chest CT scan. Biological analysis showed thrombopenia (80 G/L), lymphopenia (0.2 G/L), hypernatremia (155 mmol/L), acute kidney injury KDIGO 3 (blood creatinine: 218 µmol/L), a high level of C-reactive protein (146 mg/L) and hyperlactatemia without acidosis (arterial lactate: 2.8 mmol/L). Despite adapted treatment in the infectious disease unit with high-flow oxygen therapy, progressive hypernatremia correction and dexamethasone, the lung function worsened. Exclusive palliative care was started on 23 October, and the patient died on 29 October of a severe ARDS.

A medical autopsy was performed two days after his death. The family consented to medical autopsy and the analysis of specimens. Lungs were congestive and globally densified. The abdominal aortic lumen was occluded by a voluminous thrombus, developed on an ulcerated atheromatous plaque. Coronary and carotid arteries showed atherosclerosis, without complication.

Multiple post-mortem specimens were collected, including olfactory clefts, trachea, lungs, pleura, mediastinal lymph nodes, stomach, liver, pancreas, spleen, duodenum, transverse colon, thyroid, brain, kidneys, adrenal glands, testicles, myocardium, inferior vena cava, mesenteric artery, carotid artery, basilar artery, aorta, blood, abdominal wall muscle, healthy and damaged skin. For each anatomical localization, tissue samples were frozen for viral analyses, and other fragments were immersed in 4% formalin solution for histopathology.

### 2.2. Histopathology

After the sampling of the different organs, 4 µm-thick tissue sections were obtained and stained with hematoxylin, eosin and saffron. An immunohistochemical assay was performed with an anti-SARS-CoV-2 nucleo-capsid protein antibody (SARS-CoV Nucleocapsid Protein Antibody, rabbit polyclonal, NB100-56576 Novus Biologicals, 1/50; Flex+ Envision revelation system, Dako Omnis) on lung, myocardium, kidney, pancreas, olfactory cleft, olfactory bulb and testicle tissue samples. Positive and negative controls were used for immunohistochemistry assay validation (infected and non-infected cell lines). A positive immunostaining was defined by a granular staining of the cell cytoplasm.

### 2.3. Viral Analyses

After viral RNA extraction using Nuclisens^®^ easyMAG^®^ technology, each sample was analyzed for SARS-CoV-2 detection using RT-qPCR targeting IP2 and IP4 of the RdRp gene, according to the protocol developed by the Pasteur Institut [12]. In the case of a positive result, ultra-deep sequencing (DeepCheck^®^ Whole Genome SARS-CoV-2 Genotyping V1, ABL SA) was performed in accordance with the manufacturer recommendation on an Illumina iSeq 100 system to study the SARS-CoV-2 genetic variability according to tissue distribution. Briefly, the RT-PCR multiplex test, which consisted of 98 amplicons of approximatively 400 bp in length, spanned the entire genome of SARS-CoV-2. Subsequently, the amplicons were used for next-generation sequencing, and the downstream analysis ABL DeepCheck^®^ software was used to list SARS-CoV-2 mutations.

Moreover, to explore the evidence of an active viral replication, an additional subgenomic RNA RT-qPCR was performed, as described previously [13]. 

## 3. Results

Histopathological analyses showed the presence of severe lung diffuse alveolar damages, characterized by the presence of dystrophic pneumocytes and numerous endo-alveolar hyaline membranes, associated with a moderate inflammatory infiltrate (Figure 1A). The latter was mainly composed of macrophages and lymphocytes. A mild perivascular lymphocytic infiltrate was observed in the myocardium, without cardiomyocytes alteration. A recent mixt thrombus was seen in the abdominal aorta. The examination of the central nervous system revealed the presence of ancient frontal ischemic sequalae, without signs of recent necrosis or inflammation.

A strong staining of endoluminal macrophages associated with a mild straining of epithelial cells was found in the olfactory cleft (Figure 1B). In the lung, only endo-alveolar macrophages showed a significant positivity. In the testicles, a moderate cytoplasmic staining of seminiferous tubules cells was observed (Figure 1C). No significant staining was observed in the other organs (Figure 1D), but the analyses were limited by post-mortem autolysis (Figure 1E).

The SARS-CoV-2 genome was detected in several tissues, including the lower respiratory system, various cardio-vascular biopsies, the stomach, the pancreas, the adrenal gland mediastinal ganglion and the testicles (Figure 2, Table 1). Subgenomic RNA transcripts, in favor of an active SARS-CoV-2 replication site, were detected in all tissues positive for genome detection, except in the pancreas, aorta and vena cava. Concentrations were overall 1000-fold lower compared to genomic RNA, except in the nasopharynx (collected 9 days before death) and testicles, where they were only 10-fold lower compared to genomic RNA (Table 1), in favor of a strong viral replication.

All SARS-CoV-2 genomes detected showed that the virus belonged to the clade 20A.EU2, with the presence of the characteristic S477N and the D614G mutations on the spike protein. However, whole genome sequencing allowed the identification of numerous mutations, depending on the type of biopsy. 

Regarding the spike protein, specific mutations were observed in vascular tissues, the H655R and the V341A in vena cava (30.8% and 18.3%, respectively), as well as the E654A (44.0%), which was also poorly detected in other tissues in which the E654G was highly represented (trachea, myocardium, stomach lining, carotid and mesenteric artery) (Table 1).

For the spike protein, the Q526* mutation (ORF1a) was only detected in the mesenteric artery, and the F3271L, absent in the initial nasopharyngeal swab, was detected in pleura (14.4%), trachea (12.5%), myocardium (13.3%), stomach lining (10.5%) and testicle (9.6%).

Furthermore, the E787V mutation (ORF1b) was detected in the totality of the analyzed tissues (8.0 to 25.6% of the viral quasispecies), but was undetected in the initial nasopharyngeal swab. Various mutations were present in the stomach lining biopsy, especially the Q57H (ORF3a), reaching 98.6%, but were not detected in other tissues.

## 4. Discussion

Mutations that appear in the SARS-CoV-2 genome are responsible for the variability of viral strains circulating in the population. This variability must also be considered within viruses multiplying in the same individual as a viral quasispecies, which could determine, at least partially, the tissue tropism. The concept of genetic variability and its involvement in tissue tropism, as well as the severity of the disease, is not new, and has been previously observed for a large number of viruses including Human Immunodeficiency Virus (HIV) and Hepatitis E Virus (HEV) [14,15]. This tropism is at least partially determined by the permissiveness of the host cells viral receptor to the virus, meaning through the spike protein in the case of coronaviruses.

The ACE2 receptors are expressed in the nasopharynx but also in several other organs, making the infection possible in multiple cells, including endothelial cells, cardiomyocytes, enterocytes, parietal cells, Leydig cells, spermatogonia or Sertoli cells [16,17]. In this report, the detection of subgenomic RNA in various tissues previously positive for SARS-CoV-2 genome is an argument in favor of a local viral replication. Even though subgenomic RNA cannot be used as a current viability marker considering the persistence of RNA [18,19], the high genome concentration in tissue associated with the presence of this replicative intermediate at a high comparative level is in favor of an active SARS-CoV-2 replication at a given time. This supports the indication of exploring SARS-CoV-2 in other tissues than the respiratory tract, as previously proposed [20]. In addition to its detection outside the respiratory tract, a deep viral characterization may be relevant to better understand COVID-19 pathogeny. At the variant scale, Wagner et al. demonstrated a higher in vitro susceptibility of coronary artery endothelial cells in the case of infection with the B.1.1.7 lineage compared to wild type [21]. At the scale of viral quasispecies, Rueca et al. described the intra-host variability of SARS-CoV-2 in the upper and lower respiratory tract, with some mutations reaching 30% of the quasispecies in favor of variant compartmentalization [22]. Again, in the respiratory tract, intra-host variability was previously reported during a longitudinal follow-up of a patient infected with SARS-CoV-2 [23]. Despite these reports, the genetic variability of SARS-CoV-2 remains poorly explored outside the epidemiology context.

In this study, ultra-deep sequencing allowed the observation of several SARS-CoV-2 mutations in various proportions according to the tissue distribution. Ultra-deep sequencing has to be applied outside the respiratory tract to obtain data concerning intra-host SARS-CoV-2 quasispecies variability, differential tropism or pathogeny. When focusing on the spike protein, signature mutations associated with vascular tropism may be suggested in vena cava (i.e., V341A, E654A and H655R) (Figure 1, Table 1). These mutations have not yet been characterized in the literature; however, position 655 is also mutated in the 20J/501Y.V variant (H655Y) and has been correlated to antibody neutralization escape and to the modulation of the interaction between the spike glycoprotein and ACE2 receptors [24]. Moreover, the H655 residue is closed to the E654A mutation, and both are near the furin cleavage site of the spike protein (S1/S2 site) involved in SARS-CoV-2 cell entry. The E654A mutation suggests a significant change in the physicochemical properties of the amino acid sequence, since a glutamic acid (E) is polar and negatively charged with a higher molecular weight (MW = 128 g/mol) compared to an alanine (A), which is nonpolar and uncharged (71 g/mol). The major impact of such changes has already been described for the D614G mutation in which a polar, negatively charged aspartic acid (D) with a higher MW (114 g/mol) is substituted with a nonpolar uncharged glycine (G) with a lower MW (57 g/mol). The difference in D614G physicochemical properties has a significant impact on the three-dimensional structure of the spike protein, giving the RBD (Receptor Binding Domain) a more open conformation than the wild type strain (Wuhan-Hu-1) and, therefore, an improvement in virus interaction with the host’s cell receptors [25]. Henceforth, a potential impact on the protein functions could be suggested concerning the implication of the E654A mutation regarding its interaction with endothelial cell receptors.

Variability was not limited to the vascular system since the significant compartmentalization of viral quasi-species was also observed in other organs (i.e., the stomach lining and pleura), in association with spike mutations but also regarding other regions of the genome (ORF1a, ORF1b and ORF3a).

Additionally, viral analyses allowed the detection of SARS-CoV-2 in the testicular parenchyma. Early in the pandemic, the expression of the ACE2 receptors by germ cells, Sertoli cells and Leydig cells suggested a potential testicular reservoir for SARS-CoV-2 with possible sexual transmission [26], as well as a potential impact on male fertility [27] or even on offspring [28]. Consequently, the presence of the virus has been studied in semen from patients with infections of varying severity and expression, with conflicting results [29,30]. Further investigation of the testicular consequences of COVID-19 has highlighted that SARS-CoV-2 can directly impair male fertility through binding to the ACE2 receptor and the invasion of all testicular cell types [11]. Despite the documented infections, little is known about mutations occurring in the testes, their selective advantage and their consequences. This case report demonstrated a high level of SARS-CoV-2 replication in the testes, but the viral genome pattern was not testis specific, and to our knowledge, there is no evidence in the literature to support a link between the clinic and a testis-specific variant.

Finally, from a technical aspect, RT-qPCR appeared more sensitive than immunohistochemical assay to highlight the presence of SARS-CoV-2 in tissues. Ultra-deep sequencing, essential to study quasispecies distribution, is however, more uncertain since a low coverage may be observed for some regions of the genome (i.e., vascular samples in this study), or may even fail, despite a significant viral load in tissue (e.g., adrenal glands). The degradation of post-mortem samples may be the cause of the low sensitivity. In addition, the technique has to be improved to explore the genetic variability of SARS-CoV-2 in samples that are more complex than the classical nasopharyngeal swabs, especially in the case of low viral load, as observed here in the aorta, pancreas or mediastinal lymph nodes (i.e., Cq values ≥ 32). 

## 5. Conclusions

To summarize, SARS-CoV-2 was detected in multiple tissues collected from a death related to COVID-19. A local replication was observed in various tissues, including vessels and testicular parenchyma, leading to the relatively heterogeneous distribution of viral quasispecies associated with compartmentalization. Larger studies are now needed to explore the SARS-CoV-2 variability during the course of the infection, and in different clinical conditions, to better understand the dynamics and impact of the minority SARS-CoV-2 variants on pathogeny.

## Figures and Tables

**Figure 1 viruses-14-00515-f001:**
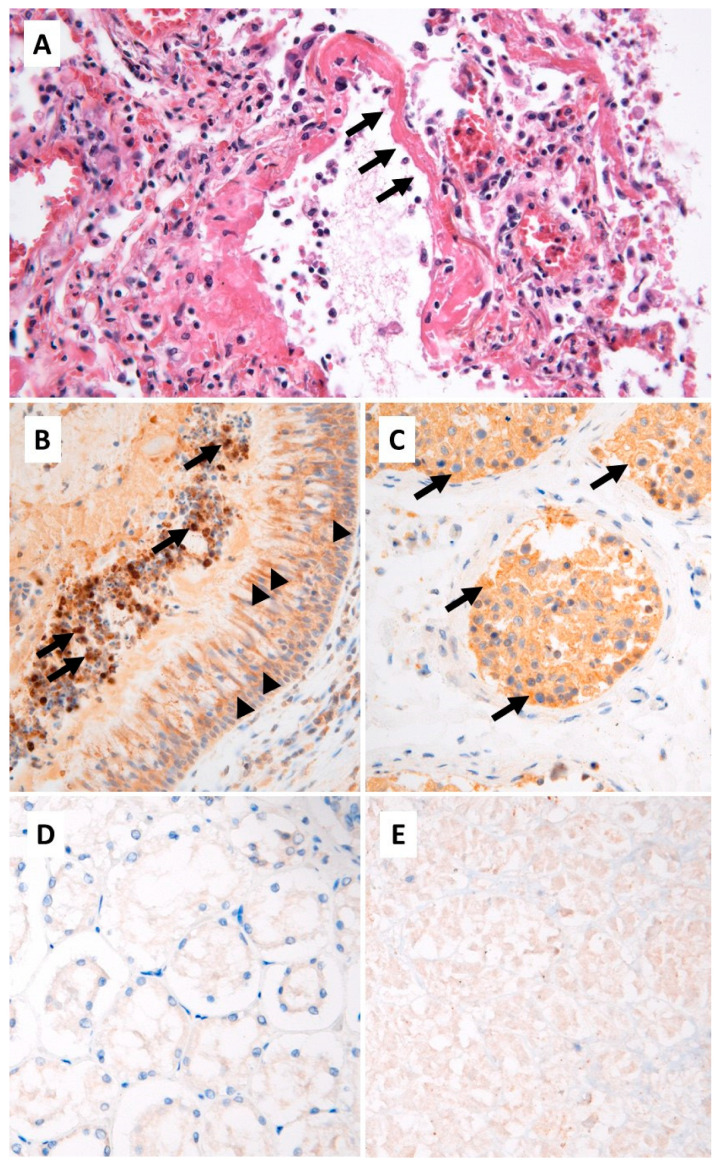
(**A**) Lung tissue, standard histology: diffuse alveolar damages with hyaline membranes (arrows) and moderate inflammatory infiltrate (hematoxylin, eosin and saffron, ×200). Immunohistochemistry, SARS-CoV-2 nucleo-capsid protein: (**B**) olfactory cleft: strong cytoplasmic staining of endoluminal macrophages (arrows) and mild staining of the epithelium (arrowheads); (**C**) testicular parenchyma: moderate cytoplasmic staining of germ cells in the seminiferous tubules (arrows); (**D**) negative staining in renal parenchyma; (**E**) non-significant weak and diffuse staining in autolytic pancreatic parenchyma (×200).

**Figure 2 viruses-14-00515-f002:**
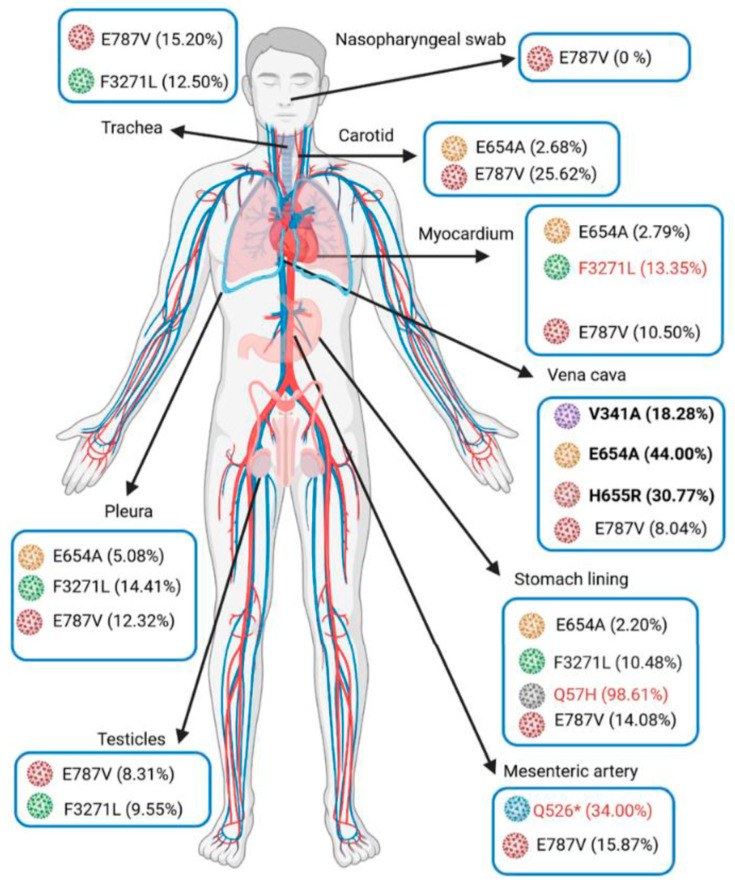
Anatomical diagram of variant distribution according to autopsied patient tissues. The amino acid colored in blue matches SARS-CoV-2 Wuhan-Hu-1 sequence (NC 045512.2: 21563-25384); amino acids with other colors match the mutated sequence (designed by biorender).

**Table 1 viruses-14-00515-t001:** Ultra-deep sequencing of SARS-CoV-2 genome isolated from post-mortem biopsies. The gradient color represents the notable proportion of mutations present inside the quasispecies. ND: not determined.

Specimen		Nasopharynx (9 Days before Death)	Trachea	Lung	Pleura	Mediastenal Lymph Nodes	Stomach	Pancreas	Adrenal Glands	Testicle	Myocardium	Inferior Vena Cava	Mesenteric Artery	Carotid Artery	Aorta
SARS-Co-2 genome detection (IP4-Cq)		27.7	24.6	23.5	24.2	32.4	23.6	32	27.3	30	25.5	28.8	22.8	20.4	31.9
Subgneomic RNA detection (Cq)		31.3	33.9	32.1	33.6	37.1	31.7	ND	34.7	32.1	34.7	ND	32.3	29.1	ND
Mutations															
ORF1a	R43H	0.00%	0.00%	ND	0.00%	failed	14.89%	failed	failed	0.00%	0.00%	ND	0.00%	0.00%	failed
	Q526*	0.00%	0.00%	0.00%	0.00%		0.00%			0.00%	0.00%	0.00%	34.00%	0.00%	
	A537V	0.00%	0.00%	0.00%	18.75%		0.00%			0.00%	9.15%	3.03%	0.00%	22.82%	
	V1968M	0.00%	0.00%	0.00%	0.00%		0.00%			0.00%	0.00%	ND	0.00%	21.66%	
	M3087I	72.73%	96.77%	100.00%	97.12%		98.39%			98.58%	97.81%	ND	ND	ND	
	F3271L	0.00%	12.50%	0.00%	14.41%		10.48%			9.55%	13.35%	0.00%	0.00%	0.00%	
	Y3300H	97.81%	96.56%	98.89%	97.58%		98.65%			98.27%	97.68%	97.54%	99.48%	99.24%	
	D3703H	0.00%	0.00%	0.00%	0.00%		0.00%			0.00%	0.00%	ND	ND	24.44%	
	G4374D	26.62%	8.41%	0.00%	0.00%		27.49%			10.25%	0.00%	ND	0.00%	0.00%	
ORF1b	A176S	100.00%	0.00%	ND	100.00%		0.00%			100.00%	92.86%	ND	ND	ND	
	P314L	100.00%	92.86%	ND	100.00%		100.00%			76.47%	100.00%	ND	ND	ND	
	V767L	98.25%	97.01%	98.40%	98.68%		98.20%			98.93%	98.71%	99.07%	99.30%	99.22%	
	E787V	0.00%	15.20%	15.13%	12.32%		14.08%			8.31%	10.50%	8.04%	15.87%	25.62%	
	K1141R	100.00%	92.06%	100.00%	94.44%		100.00%			89.47%	96.00%	ND	100.00%	98.80%	
	E1184D	90.91%	100.00%	100.00%	84.00%		97.01%			93.75%	92.93%	ND	100.00%	100.00%	
Spike	V341A	0.00%	0.00%	0.00%	0.00%		0.00%			0.00%	0.00%	18.28%	0.00%	0.00%	
	S477N	100.00%	98.76%	100.00%	97.98%		98.94%			100.00%	98.98%	100.00%	97.30%	100.00%	
	D614G	98.82%	98.29%	98.77%	98.98%		99.29%			98.50%	99.02%	98.98%	99.60%	99.51%	
	E654A	0.00%	0.00%	0.00%	5.08%		2.20%			0.00%	2.79%	44.00%	0.00%	2.68%	
	E654G	9.14%	16.30%	0.00%	0.00%		18.18%			7.05%	15.08%	0.00%	14.35%	18.21%	
	H655R	0.00%	0.00%	0.00%	0.00%		0.00%			0.00%	0.00%	30.77%	0.00%	0.00%	
	I1172T	0.00%	0.00%	0.00%	10.91%		10.82%			10.46%	8.55%	0.00%	0.00%	0.00%	
ORF3a	Q57H	0.00%	0.00%	0.00%	0.00%		98.61%			0.00%	0.00%	0.00%	0.00%	0.00%	
	P207S	ND	ND	ND	ND		0.00%			ND	ND	ND	25.00%	ND	
N	K169N	0.00%	0.00%	0.00%	13.10%		3.32%			5.27%	2.80%	0.00%	0.00%	0.00%	
	M234I	94.12%	97.33%	81.25%	96.64%		99.17%			98.71%	98.95%	100.00%	98.61%	99.61%	
	K256*	ND	0.00%	10.27%	10.53%		10.18%			11.81%	7.76%	13.51%	0.00%	8.76%	
	A376T	100.00%	100.00%	100.00%	98.00%		98.96%			100.00%	98.98%	100.00%	98.18%	99.25%	

## Data Availability

Not applicable.

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
