# Peer review of "Multiorgan and Vascular Tropism of SARS-CoV-2"

_viruses, 2022, doi:10.3390/v14030515_

Round 1

Reviewer 1 Report

This is an interesting case report addressing the SARS-CoV2 tropism in different tissues. The authors found that the virus can be detected in many different tissues, showing signs of active replication in some of them and, showing mutations according to different tissue tropism.

The case report is mostly well written, and I think it would be of interest for publication. I have the following comments:

In line 120, I don’t understand the phrase “in the olfactory cleft, it showed a strong…” I believe what the authors mean is that “A strong staining of endoluminal macrophages associated with a mild straining of epithelial cells was found in the olfactory cleft”

The authors should discuss the reasons why they did not determine the mutations of the SARS-CoV-2 genome in certain organs as shown in table 1 and why their technique failed in others. These factors should be acknowledged as limitations of the paper.

Author Response

All the proposals have been considered.

Reviewer 1:

In line 120, I don’t understand the phrase “in the olfactory cleft, it showed a strong…” I believe what the authors mean is that “A strong staining of endoluminal macrophages associated with a mild straining of epithelial cells was found in the olfactory cleft”

Change has been made L129-130.

The authors should discuss the reasons why they did not determine the mutations of the SARS-CoV-2 genome in certain organs as shown in table 1 and why their technique failed in others. These factors should be acknowledged as limitations of the paper.

Thanks for this proposition, limitations have been added in the discussion line 274-282.

Reviewer 2 Report

Here are some of my criticisms: 

  1. The detail of ultra-deep sequencing is not provided. Please describe more about how this method was conducted.
  2. Table 1 is comprehensive, while the characters are too small to be seen. Please enlarge the size of the whole table. 
  3. Mutations of ORF1a, 1b Spike and N genes were shown in Table 1. Are there any other ORFs with mutation found?
  4. On the histopathological images, please define (or indicate which part is e.g. the macrophages or other types of cells) clearly.
  5. How do the authors define the staining of significance, positive or negative with CoV-2 virus? Can the authors show some examples of negative staining and some with post-mortem autolysis? Also please explain why some data in Table 1 are "failed"? Is that also due to "post-mortem autolysis"?
  6. Please describe the result of Figure 1 in the order of the A (lung), B (olfactory cleft), and C (testicular parenchyma). And why there's no data of olfactory cleft in the Table 1?
  7. The major finding of this case study is the 3 mutations (V341A, E654A and H655R) existing only in inferior vena cava. This is a nice point worthy further investigation. The authors claimed this is vascular tropism. While in other regions of the vascular system tested (e.g. Mesenteric artery, Carotid artery, and Myocardium), no such mutations were found. Pleas explain this or describe more about other mutations existed in the other vascular regions.

Author Response

 All the proposals have been considered.

Reviewer 2:

  1. The detail of ultra-deep sequencing is not provided. Please describe more about how this method was conducted.

Precisions have been added L103-112.

  1. Table 1 is comprehensive, while the characters are too small to be seen. Please enlarge the size of the whole table.

The representation of Table 1 in portrait orientation lead to a better comprehension. Change have been proposed page 6.

  1. Mutations of ORF1a, 1b Spike and N genes were shown in Table 1. Are there any other ORFs with mutation found?

All information available concerning the mutations detected are presented in Table 1.

  1. On the histopathological images, please define (or indicate which part is e.g. the macrophages or other types of cells) clearly.

Arrows and arrowheads have been added in the figure 1 and legend figure.

  1. How do the authors define the staining of significance, positive or negative with CoV-2 virus? Can the authors show some examples of negative staining and some with post-mortem autolysis? Also please explain why some data in Table 1 are "failed"? Is that also due to "post-mortem autolysis"?

Positive and negative controls have been used for immunohistochemistry assay validation (infected and non-infected cell lines) (added in material and methods, page 3, line 97-98).

Positive immunostaining for SARS-CoV-2 was defined by a granular cytoplasm staining (added in material and methods, page 3, line 99-100).

Following the reviewer’s suggestion, we added an example of negative staining (renal parenchyma, Figure 1 D) and an example of nonspecific staining in autolytic pancreatic tissue (Figure 1 E). We also improved the quality of Figure 1 C.

Thanks for this proposition, explanations concerning failed data have been added in the discussion line 274-282.

  1. Please describe the result of Figure 1 in the order of the A (lung), B (olfactory cleft), and C (testicular parenchyma). And why there's no data of olfactory cleft in the Table 1?

References to Figure 1 A, 1 B and 1 C have been added in the results section, page 3, line 122, 130, 132; the figure 1 legend has been completed.

Unfortunately, virological analyses have not been performed on olfactory cleft due to the lack of sample.

  1. The major finding of this case study is the 3 mutations (V341A, E654A and H655R) existing only in inferior vena cava. This is a nice point worthy further investigation. The authors claimed this is vascular tropism. While in other regions of the vascular system tested (e.g. Mesenteric artery, Carotid artery, and Myocardium), no such mutations were found. Please explain this or describe more about other mutations existed in the other vascular regions.

The identification of mutations only present in viruses isolated from vena cava is actually an important finding of this case report, although absent from mesenteric or carotid artery. In the same way, others mutations have been identified in high proportion in viruses isolated from mesenteric (Q526*) or carotid artery (A537V, V1968M) but absent or in low prevalence in inferior vena cava.

Little is known concerning the tropism of SARS-CoV-2 according to tissues as well as concerning the role of these mutations, even if some mechanistic properties could be suggested (L240-253). Further investigations have now to be performed following the demonstration here of the intra-host variability of SARS-CoV-2, especially regarding the vascular tissue.